# Limited Cheese Intake Paradigm Replaces Patterns of Behavioral Disorders in Experimental PTSD: Focus on Resveratrol Supplementation

**DOI:** 10.3390/ijms241814343

**Published:** 2023-09-20

**Authors:** Vadim E. Tseilikman, Vladislav A. Shatilov, Maxim S. Zhukov, Irina A. Buksha, Alexandr E. Epitashvily, Ilya A. Lipatov, Maxim R. Aristov, Alexandr G. Koshelev, Marina N. Karpenko, Dmitrii S. Traktirov, Viktoriya A. Maistrenko, Mustapha Kamel, Alexey V. Buhler, Elena G. Kovaleva, Tatyana S. Kalinina, Anton A. Pashkov, Vadim V. Kon’kov, Jurica Novak, Olga B. Tseilikman

**Affiliations:** 1Scientific and Educational Center ‘Biomedical Technologies’, School of Medical Biology, South Ural State University, 454080 Chelyabinsk, Russia; vlad.shatilov.2018@mail.ru (V.A.S.); zhukovmax@mail.ru (M.S.Z.); maksimaristov04@mail.ru (M.R.A.); sch_viktoriya@mail.ru (V.A.M.); zellist@mail.ru (A.V.B.);; 2Faculty of Fundamental Medicine, Chelyabinsk State University, 454001 Chelyabinsk, Russia; bukshairina2017@gmail.com (I.A.B.); ilyuha861@gmail.com (I.A.L.); sanatorium57@yandex.ru (A.G.K.); 3Pavlov Department of Physiology, Institute of Experimental Medicine, 197376 Saint Petersburg, Russia; mnkarpenko@mail.ru (M.N.K.); ds.tractirov@gmail.com (D.S.T.); 4Research, Educational and Innovative Center of Chemical and Pharmaceutical Technologies Chemical Technology Institute, Ural Federal University Named after the First President of Russia B.N. Yeltsin, 620002 Ekaterinburg, Russia; ur.ufru@kovaleva.g.e; 5Institute of Cytology and Genetics, Siberian Branch of the Russian Academy of Science, 630090 Novosibirsk, Russia; smartkalin@gmail.com; 6Federal Neurosurgical Center, 630048 Novosibirsk, Russia; pashkov-anton@mail.ru; 7Department of Data Collection and Processing Systems, Novosibirsk State Technical University, 630087 Novosibirsk, Russia; 8Zelman Institute of Medicine and Psychology, Novosibirsk State University, 630090 Novosibirsk, Russia; 9Department of Biotechnology, University of Rijeka, 51000 Rijeka, Croatia; 10Center for Artificial Intelligence and Cyber Security, University of Rijeka, 51000 Rijeka, Croatia

**Keywords:** resveratrol, PTSD, anxiety, freezing, monoamines oxidize-A, brain-derived neurotrophic factor, dopamine, serotonin

## Abstract

Currently, the efficacy of drug therapy for post-traumatic stress disorder or PTSD leaves much to be desired, making nutraceutical support a promising avenue for treatment. Recent research has identified the protective effects of resveratrol in PTSD. Here, we tested the behavioral and neurobiological effects of combining cheese consumption with resveratrol supplements in an experimental PTSD model. Using the elevated plus maze test, we observed that cheese intake resulted in a shift from anxiety-like behavior to depressive behavior, evident in increased freezing acts. However, no significant changes in the anxiety index value were observed. Interestingly, supplementation with cheese and resveratrol only led to the elimination of freezing behavior in half of the PTSD rats. We further segregated the rats into two groups based on freezing behavior: Freezing^+^ and Freezing^0^ phenotypes. Resveratrol ameliorated the abnormalities in Monoamine Oxidize -A and Brain-Derived Neurotrophic Factor gene expression in the hippocampus, but only in the Freezing^0^ rats. Moreover, a negative correlation was found between the number of freezing acts and the levels of Monoamine Oxidize-A and Brain-Derived Neurotrophic Factor mRNAs in the hippocampus. The study results show promise for resveratrol supplementation in PTSD treatment. Further research is warranted to better understand the underlying mechanisms and optimize the potential benefits of resveratrol supplementation for PTSD.

## 1. Introduction

Post-traumatic stress disorder (PTSD) is a stress-related mental illness triggered by overwhelming stress. It can often be complicated by somatic diseases such as ischemic heart illness, atherosclerosis, essential arterial hypertension, stroke, diabetes mellitus, metabolic syndrome, immunopathology, and even cancers [1,2,3,4,5,6,7]. However, the primary symptoms of PTSD manifest as behavioral features, including hyperarousal, nightmares, flashbacks, anger, aggressiveness, fear, and anxiety [8,9,10]. The development of PTSD is associated with provoked stress time-dependent sensitization (TDS), which involves a series of neurobiological events with lasting consequences, including alterations in gene expression [11,12,13,14,15,16].

The hippocampus, a key brain region, plays a crucial role in the pathogenesis of PTSD. Studies involving PTSD patients have consistently shown a reduction in hippocampus volume [17,18]. Furthermore, exposure to repeated emotional stressors, such as those experienced in PTSD, leads to a decline in the number of dendrites in hippocampal neurons and triggers anxiety-like behavior, such as agoraphobia [19]. Neurochemical profiling has revealed increased turnover rates of dopamine (DA) and serotonin (5-HT) in the hippocampus during exposure to predator scent in animal models of PTSD, as well as decreased neurotransmitter concentrations in this brain region [20,21,22]. Additionally, an increase in hippocampal norepinephrine (NE) concentration has been observed. Interestingly, the activity and expression of monoamine oxidases (MAO), enzymes responsible for the oxidative deamination of monoamines, were found to be decreased in the hippocampus [23].

Nowadays, selective serotonin reuptake inhibitors (SSRIs) are widely used in the treatment of PTSD, with the main target being the 5-HT transporter (SERT). Experimental PTSD models, such as the single prolonged test, have revealed that SSRIs can correct both behavioral and neurochemical abnormalities in the hippocampus [24,25]. It is suggested that the efficacy of SSRIs in addressing behavioral disorders is directly linked to their ability to reverse abnormalities in hippocampal neurotransmitter metabolism [26]. However, SSRIs are associated with multiple side effects, which limits their effectiveness in managing PTSD [27].

Interestingly, *trans*-resveratrol (RES), a natural polyphenol belonging to the phytoalexins similar to SSRIs, has been found to downregulate the mRNA expression of 5-HT transporter in the hippocampus [28]. Recent studies have shown that RES can alleviate anxiety-like behavior and decrease brain-derived neurotrophic factor (BDNF) levels, which are typically decreased in stressed rats in experimental PTSD models [29,30]. It is worth noting that PTSD-resilient rats exhibit increased hippocampal BDNF mRNA levels along with simultaneous increases in DOP (DAT) transporter mRNA levels in the hippocampus [30].

Nowadays, the multifaceted physiological benefits of resveratrol have garnered considerable attention, showcasing its potential to enhance healthspan across diverse experimental models of various behavioral disorders. Clinical trials have also lent support to the ongoing exploration of resveratrol treatment in human contexts [31]. Notably, *trans*-resveratrol supplementation has demonstrated its ability to enhance cognitive function in the elderly [32]. However, the therapeutic application of RES is limited due to its high biotransformation rates by liver microsomal enzymes [33,34,35]. Nevertheless, certain microorganisms can convert RES into more stable metabolites, such as dihydroresveratrol (DHR), resveratrol-sulfate, and resveratrol-glucuronide [36]. Therefore, supplementing RES in products that incorporate probiotics, such as yogurt, cheese, juices, and jams, might be beneficial. Cheese consumption is considered a highly palatable and calorically dense food often sought for comfort during stress [37].

Recently, it has been shown that a limited cheese intake diet (LCI) reduces behavioral and hypothalamic-pituitary-adrenocortical (HPA) axis responses to stress in rats subjected to a restraint stress paradigm [37]. Additionally, LCI increases neuronal activity in the nucleus accumbens (NAc), where the body of dopaminergic neurons is located [37]. This influence of LCI on the NAc resembles the effects of other types of natural and pharmacological rewards. Therefore, we propose that using cheese as a vehicle for RES supplementation could be a reasonable approach for nutraceutical support in managing PTSD.

In this study, we aim to evaluate the impact of RES supplementation on the protective effects of LCI against PTSD.

## 2. Results

Protocol #1 Behavioral activity, hippocampal monoamine metabolism, and gene expression of PS (Predator Stress) rats.

To assess the impact of predator stress (PS) on behavioral activity, hippocampal monoamine metabolism, and gene expression of hippocampal metabolic enzymes and BDNF, we initiated a 10-day exposure of rats to PS. Following a resting period of 14 days, we conducted an elevated plus maze (EPM) test to evaluate the anxiety levels of each rat (Figure 1).

### 2.1. Behavioral Activity in the Elevated Plus Maze Test

Table 1 shows the results of the behavioral test for the control group rats and PS rats with a cheese-free ration. The Mann–Whitney test showed that the experimental PS group was significantly affected in behavior evaluation via the elevated plus maze test.

For PS rats, the time spent in the closed arms of the maze was 10% longer than for control rats (*p* < 0.001). Meanwhile, PS rats spent 51% less time in the open arms than did the control rats (*p* < 0.001). The number of entries into the open arms by the PS rats was 50% fewer than for the control rats (*p* < 0.001). The number of entries into the closed arms was 31% less than for the control rats (*p* < 0.001). The anxiety index (AI) of the PS rats was 11% higher than that of the control rats (*p* < 0.001). PS did not affect freezing in the EPM test.

### 2.2. Neurochemical Abnormalities in the Hippocampus of PS Rats

#### Impact of PS on Hippocampal Monoamine Concentrations and Their Metabolites

Figure 2 illustrates the impact of PS on the concentrations of monoamines (a), their metabolites (b), and turnover rates (c) in the hippocampus. The Mann–Whitney test revealed significant differences between PS rats and control rats in hippocampal norepinephrine (NA), dopamine (DA), and 5-hydroxytryptamine (5-HT) concentrations. In PS rats, hippocampal NA concentration was 37% (*p* = 0.013) higher than in the control group, while hippocampal DA concentration was 46% (*p* = 0.002) lower, and hippocampal 5-HT concentration was 20% (*p* = 0.037) lower (Figure 2A). Additionally, hippocampal homovanillic acid (HVA) concentration, a DA metabolite, was 46% (*p* = 0.002) lower in PS rats compared to the control group. However, PS did not significantly affect the concentrations of other DA metabolites, such as dihydroxyphenylacetate (DOPAC), or 5-HT metabolites, such as 5-hydroxyindolacetic acid (5HIAA), in the hippocampus (Figure 2B). Furthermore, 5-HT turnover rates in PS rats were 222% (*p* = 0.044) higher than in the control group, while DA turnover rates were not significantly affected by PS (Figure 2C).

The Mann–Whitney test revealed a significant difference in hippocampal MAO-A activity between control and PS rats (Figure 2D). Specifically, in PS rats, hippocampal MAO-A activity was 52% lower than in the control group (*p* = 0.0015). Furthermore, a positive correlation was observed between MAO-A activity and 5-HT turnover rate in the hippocampus (r = 0.63; *p* < 0.05). On the other hand, PS did not have any significant effect on hippocampal MAO-B activity. Interestingly, there was a positive correlation between MAO-B activity and hippocampal DA concentration (r = 0.7, *p* < 0.05), as well as its metabolite HVA (r = 0.63, *p* < 0.05).

### 2.3. The Impact of PS on Hippocampal DAT, MAO-A, MAO-B, COMT, and BDNF mRNA Expression

In Figure 3, it can be observed that the content of MAO-A mRNA in the hippocampus of PS rats decreased significantly by half (*p* = 0.032), while the content of COMT mRNA doubled (*p* = 0.045) compared to the control group (Figure 3A). However, the Mann–Whitney test did not reveal significant differences in the content of MAO-B (*p* = 0.52), DAT (*p* = 0.132), and BDNF (*p* = 0.07) mRNAs (Figure 3A,B). Interestingly, a positive correlation (r = 0.57) was found between the content of MAO-A mRNA and the time spent in the open arms.

Protocol#2 Effect of LCI on behavioral disorders, neurochemical abnormalities and altered gene expression in the hippocampus of PS rats.

To evaluate the effect of LCI on behavioral activity and monoamine metabolism in the hippocampus of PS rats, we first exposed the rats to PS for 10 days (Figure 4). The PS-subjected rats were then divided into two groups: PS^#^ rats, which received cheese without RES supplementation, and RES^+^ rats, which received cheese with RES supplement. Similarly, the control group continued to receive cheese without any PS exposure. After a 14 days’ rest period for both groups of stressed rats, was performed an elevated plus maze (EPM) test to assess their behavior.

### 2.4. Effect of LCI on Behavioral Activity of PS Rats in the EPM Test

#### 2.4.1. LCI Prevented the Development of Anxiety-like Behavior and Simultaneously Increased Freezing in PS^#^ and RES^+^ Rats

The impact of cheese consumption on behavioral activity is depicted in Figure 5. When PS rats (PS^#^ group) were fed cheese, it prevented anxiety-like behavior, as shown in Figure 5A–F, while simultaneously enhancing freezing behavior (Figure 5E). One-way ANOVA did not reveal significant effects of cheese consumption on the time spent in the open and closed arms (F2,26 = 1.01; *p* = 0.35 for both), the number of entries into the open (F2,26 = 0.52; *p* = 0.59) and closed arms (F2,26 = 0.69; *p* = 0.58), and the AI (F2,26 = 1.92; *p* = 0.16) in the PS^#^ rats. However, in this group, freezing behavior was 3100% higher than that of the control rats (*p* < 0.001).

#### 2.4.2. LCI Segregated RES^+^ Rats into Two Behavioral Phenotypes Based on Their Freezing Levels

Analysis of the entire sample of PS^#^ rats revealed that dietary RES supplementation (RES^+^ rats) had no effect on freezing behavior (Figure 6). However, in 50% of RES^+^ rats, freezing behavior completely disappeared. Thus, RES^+^ rats were classified into two phenotypes. In the first phenotype (Freezing^+^ rats), freezing behavior was not affected. In the second phenotype (Freezing^0^ rats), freezing behavior disappeared after cheese consumption. One-way ANOVA showed significant effects of cheese consumption on the time spent in both the open and closed arms (in both cases F3,25 = 4.38; *p* = 0.35), on the anxiety index (F3,25 = 4.28; *p* = 0.016), and on freezing behavior (F3,25 = 9.17; *p* = 0.0006) in the RES^+^ rats.

The data presented in Figure 6 suggest an anxiolytic effect in Freezing^+^ rats compared to Freezing^0^ rats. For Freezing^+^ rats, the time spent in the closed arms was 61% (*p* = 0.005) shorter than in Freezing^0^ rats and 53% (*p* = 0.025) lower compared to the PS group. As expected from the time spent in the closed arms, Freezing^+^ rats spent more time in the open arms than Freezing^0^ rats (−70%, *p* = 0.009). Moreover, in Freezing^+^ rats, the time spent in the open arms was 330% (*p* = 0.007) longer, and the time spent in the closed arms was 60% (*p* = 0.007) shorter than in the control group. Consequently, the AI value was 62% (*p* = 0.008) lower than in the control group. In contrast, in Freezing^0^ rats, no significant differences were found compared to the control group and the PS rats in the EPM test values.

### 2.5. LCI Reduces Hippocampal MAO-A/MAO-B Activities, Prevents Abnormalities in DA Metabolism in the Hippocampus of PS^#^ Rats, but Cannot Correct Abnormalities in 5-HT Metabolism

The one-way ANOVA test revealed significant effects of PS on NA (F2,26 = 19.86, *p* = 0.0006), DA (F2,26 = 5.06, *p* = 0.01), 5-HT (F2,26 = 19.13, *p* = 0.0006), DOPAC (F2,26 = 21.35, *p* = 0.0004), and 5HIAA (F2,26 = 66.27, *p* = 0.00002) concentrations in cheese-fed rats. However, PS did not significantly affect HVA concentration (F2,26 = 1.44, *p* = 0.25) (Figure 7). Additionally, the one-way ANOVA test revealed significant effects of PS on DA (F2,26 = 21.89, *p* = 0.0003) and 5-HT (F2,26 = 58.64; *p* = 0.35) turnover rates in cheese-fed rats.

PS significantly increased NA and DA concentrations in the hippocampus. Post-hoc analysis revealed that hippocampal NA concentration was 75% higher (*p* = 0.0001), and hippocampal DA was twice as high as in control rats (*p* = 0.013). However, 5-HT concentration was 39% lower in PS rats compared to the control group (*p* = 0.0031). A significant negative correlation was found between hippocampal DA concentration and freezing behavior (r = −0.52; *p* < 0.05).

PS did not affect hippocampal DOPAC concentration in cheese-fed rats. However, the hippocampal 5HIAA concentration in PS^#^ rats was 23% lower than in the control group (*p* = 0.032). Interestingly, a significant positive correlation was found between hippocampal 5HIAA concentration and freezing behavior (r = 0.79, *p* < 0.05).

Furthermore, PS markedly increased 5-HT turnover rate (Figure 8), which was 25% higher than in control rats (*p* = 0.025).

The one-way ANOVA test showed a significant effect of LCI on the hippocampal MAO-A (F3,24 = 6.91; *p* = 0.016) and MAO-B activities (F3,24 = 3.38; *p* = 0.034) in PS^#^ rats (Figure 8). Post-hoc analysis revealed that in PS^#^ rats, MAO-A activity decreased by 59% (*p* = 0.015) compared to the control group. Additionally, it was found that in PS^#^ rats, MAO-B activity decreased by 22% (*p* = 0.019) compared to the control animals. Moreover, positive correlations were found between hippocampal MAO-A activity and AI value (r = 0.69, *p* < 0.05), as well as negative correlations between hippocampal MAO-A activity and freezing (r = −0.89, *p* < 0.05) in PS^#^ rats.

### 2.6. Effects of RES Supplementation and LCI on Hippocampal MAO-A/MAO-B Activities and Monoamine Abnormalities in PS-Subjected Rats

RES supplementation has effectively halted the increase in hippocampal DA in PS^#^ rats (Figure 9A). The DA concentration in RES^+^ rats was 41% (*p* = 0.035) lower than in PS^#^ rats. However, daily intake of cheese with RES supplementation did not alleviate the increase in NA nor the decrease in 5-HT in the hippocampus of PS rats. In RES^+^ rats, hippocampal NA concentration was 69% (*p* = 0.005) higher, while hippocampal 5-HT concentration was 35% (*p* = 0.015) lower than in the control group. There were no significant differences between PS^#^ and RES^+^ rats in the NA (*p* = 0.34) and DA (*p* = 0.28) concentrations. However, in RES^+^ rats, there was a tenfold (*p* = 0.0005) decrease in DOPAC concentration compared to both PS^#^ rats and the control group (*p* = 0.0005). Hippocampal DA turnover in RES^+^ rats was 81% (*p* = 0.0005) lower than in the PS^#^ group and 80% (*p* = 0.0005) lower than in the control group.

Hippocampal 5-HIAA concentration in RES^+^ rats was 93% (*p* = 0.0003) lower than in PS^#^ rats and 95% (*p* = 0.0003) lower than in control rats. Additionally, in RES^+^ rats, a reduction in hippocampal 5-HT turnover rate was also indicated. Specifically, 5-HT turnover rate in RES^+^ rats was 92% (*p* = 0.0003) lower compared to PS^#^ rats and was tenfold (*p* = 0.0005) lower than in the control group. RES supplementation reversed the diminishing of MAO-A activity in the hippocampus, but only in Freezing^0^ rats. Figure 10 shows that in this behavioral phenotype, MAO-A activity has been completely recovered and did not differ from the control group. In freezing^0^ rats, hippocampal MAO-A activity was 54% (*p* = 0.012) higher than in PS^#^ rats. On the other hand, Freezing^+^ rats had the lowest levels of hippocampal MAO-A activity, with activity being 45% (*p* = 0.045) less than in PS^#^ rats (Figure 8B).

### 2.7. LCI Enhanced the Impact of PS on the Expression of Genes Related to Monoamine Metabolism, as Well as on the Genes of DAT and BDNF in the Hippocampus

One-way ANOVA revealed a significant influence of LCI on the mRNA levels of MAO-A (F2,20 = 30.2; *p* < 0.0001), MAO-B (F2,21 = 17.21; *p* < 0.0001), COMT (F2,21 = 7.1; *p* = 0.004), DAT (F2,20 = 3.72; *p* = 0.03), and BDNF (F2,21 = 3.72; *p* = 0.041) in the hippocampus of PS-exposed rats (Figure 11). In PS^#^ rats, the hippocampal MAO-A mRNA level was 77% (*p* = 0.0037) lower than in control rats, while the MAO-B mRNA level diminished by 68% (*p* = 0.0007) compared to the control group. Additionally, hippocampal COMT mRNA levels were threefold higher (*p* = 0.005), and the DAT mRNA level was twice as high (*p* = 0.005) as in the control group. The hippocampal BDNF mRNA level was also twice as high (*p* = 0.005) compared to the control group.

RES supplementation exacerbated the down-regulation effect of LCI on hippocampal MAO-A mRNA in PS^#^ rats. In RES^+^ rats, hippocampal MAO-A mRNA was twofold lower (*p* = 0.0037) than in the control group. However, RES^+^ supplementation did not affect the LCI-induced changes in the mRNA levels of MAO-B, COMT, DAT, and BDNF in the hippocampus of PS^#^ rats.

### 2.8. Divergent Effects of RES Supplementation on Hippocampal Gene Expression in Two Behavioral Phenotypes of RES^+^ Rats

One-way ANOVA revealed significant differences in hippocampal gene expression among PS^#^ rats with RES supplementation. Specifically, there were significant effects of RES on hippocampal MAO-A (F3,22 = 39.2; *p* < 0.0001), hippocampal MAO-B (F3,22 = 14.5; *p* < 0.0001), hippocampal COMT (F3,22 = 6.3; *p* = 0.01), hippocampal DAT (F3,22 = 7.62; *p* = 0.01), and hippocampal BDNF (F3,22 = 8.62; *p* = 0.0005) mRNA levels.

Among the different behavioral phenotypes, Freezing^+^ rats exhibited the lowest MAO-A levels compared to all other experimental groups. Specifically, MAO-A mRNA level in Freezing^0^ rats was 223% higher than in Freezing^+^ rats (*p* = 0.00017), and MAO-A mRNA levels in PS^#^ rats were three times higher than in Freezing^+^ rats (*p* = 0.00011). Notably, MAO-A mRNA levels in Freezing^0^ rats were 37% lower than in PS^#^ rats (*p* = 0.01).

In Freezing^+^ rats, the COMT mRNA level was three times higher than in control rats (*p* = 0.0019), and the LCI-induced effect on DAT and BDNF gene expression was reversed. Specifically, hippocampal DAT mRNA levels were two and a half times lower in Freezing^+^ rats compared to Freezing^0^ rats (*p* = 0.00024) and 75% lower than in PS rats (*p* = 0.0001) (Figure 12). Likewise, hippocampal BDNF mRNA level in Freezing^+^ rats was 79% lower than in Freezing^0^ rats (*p* = 0.00017) and 84% lower than in PS rats (*p* = 0.00024).

## 3. Discussion

The main findings of our study are as follows. Firstly, an important consequence of cheese intake after PS exposure was the alteration of behavioral patterns, as revealed in the EPM test (Figure 13). Instead of the expected time-dependent development of anxiety-like behavior in PS^#^ rats, we observed an enhancement of fear behavior that resulted in freezing increase (Figure 14).

Secondly, dietary RES supplementation in cheese altered the behavioral response of PS^#^ rats (Figure 15). RES^+^ rats were classified into two phenotypes, namely Freezing^+^ and Freezing^0^ rats. Freezing^+^ rats exhibited a more defined anxiolytic response than Freezing^0^ rats (Figure 16).

Thirdly, cheese intake post cessation of PS reversed some neurochemical alterations in the hippocampus. Instead of the hippocampal DA reduction inherent to PS rats, we observed a prominent rise in hippocampal DA concentrations after cheese intake.

Fourthly, dietary RES^+^ supplementation reversed the 5-HT turnover rate. While PS rats showed an elevation in hippocampal 5-HT turnover rate, RES^+^ rats exhibited a notable reduction in this parameter.

Lastly, Freezing^0^ rats were characterized by higher hippocampal MAO-A activity compared to Freezing^+^ rats.

Our data suggests that limited cheese intake prevented the development of anxiety-like behavior in the experimental PTSD model. Notably, cheese is considered a palatable food and is sometimes referred to as “comfort food” in colloquial terms [37,38]. Therefore, cheese, as a functional food, may play a more prominent role in the future. In this context, our results may be useful for further trials of functional cheese products for humans. Importantly, our findings are consistent with the results of other studies. Recently, it has been shown that a history of limited cheese intake (LCI) reduced HPA axis responses to acute restraint and physiological hypoxia stressors. Specifically, LCI promoted FosB/deltaFosB expression [37], a protein associated with repeated or chronic neuronal activation in the nucleus accumbens (NAc) [37]. These results suggest that palatable foods, regardless of their sugar/carbohydrate composition, can provide stress relief and support the idea that food reward itself contributes to stress mitigation [39].

It is well-known that the NAc is a dopaminergic nucleus. In our experimental model of PTSD, we previously observed a decrease in DA concentration in some brain regions, such as the hippocampus and hypothalamus [20,40]. Recently, it has been demonstrated that decreased DA concentration in the brain in PTSD-susceptible rats is associated with a reduction in cerebral blood flow [40]. Low DA levels are generally associated with an increased risk for PTSD, and this state is genetically predetermined [41]. In clinical studies, a so-called “pro-dopamine regulator,” nutraceutical KB220Z, was successfully used to manage PTSD-related recurrent distressing nightmares [42]. In our previous study involving PS exposure [20], cerebral DA concentrations were also decreased in PS-exposed rats. However, in this study, we demonstrate for the first time that post-stress, DA concentrations increased in rats resilient to PS. In chronic stress, decreased DA in the brain is associated with local activation and degeneration of microglia, while the administration of exogenous DA maintains autoregulation of damaged structures and prevents their necrosis [43].

In previous studies, DA has been associated with PTSD resilience. The marked increase in DF concentration in PS^#^ rats might be explained by the high concentrations of trace amines in cheese. A substantial body of evidence suggests that trace amines play significant roles in coordinating biogenic amine-based synaptic physiology. At high concentrations, they exhibit well-characterized presynaptic “amphetamine-like” effects on catecholamines and indolamines release, reuptake, and biosynthesis. At lower concentrations, they possess postsynaptic modulatory effects that potentiate the activity of DA [44]. Regarding milk-based fermented foods, cheese is the main product likely to contain potentially harmful levels of tyramine [45]. Tyramine, along with other trace amines, might be regulating neuronal DA effects by intervening in DA reuptake. Notably, tyramine potentially can be converted to DA via the CYP2D6-dependent pathway. These findings add to the understanding of the complex interactions between cheese intake, monoamine neurotransmission, and stress response in the context of PTSD, and may provide valuable insights for future research and potential therapeutic approaches.

DA is metabolized by two enzymes, MAO-A and MAO-B, as well as by COMT. DOPAC is a DA metabolite dependent on MAO activity, while HVA is a metabolite dependent on COMT. MAO-A and MAO-B are localized in the mitochondria of presynaptic neurons, where they metabolize DA that is taken back into the presynaptic neuron through DAT (dopamine transporter). On the other hand, COMT is associated with the postsynaptic neuron and directly metabolizes DA in the postsynaptic membrane. Thus, there are two pathways for DA metabolism: pathway 1, where COMT activation predominates, and pathway 2, where MAO activation predominates.

In the PS rats, we observed a decrease in MAO-A mRNA content, while there was a slight tendency towards an increase in COMT mRNA content. Conversely, cheese intake led to a significant increase in COMT mRNA content and a decrease in MAO-A and MAO-B mRNA content in PS^#^ rats. At the same time, we observed an elevation of DAT mRNA content in PS^#^ rats. Taken together, these data suggest that cheese intake leads to an enhancement of DA reuptake. However, the low levels of MAO-A expression and MAO-A activity indicate the presence of an opportunity to store DA in vesicles for subsequent release into the synaptic gap with subsequent activation of the COMT pathway.

Importantly, we also found an increase in BDNF mRNA content in PS^#^ rats. BDNF is a major neuroplasticity factor, and reports on the trigger role of the BDNF-TrkB signaling pathway in the formation of vesicles with DA looks promising [46].

Overall, the data obtained in this study align well with our previous reports on the association of DA concentration in the hippocampus with PTSD resilience [30]. However, in PS^#^ rats, cheese intake prevented the development of anxiety-like behavior but did not limit freezing. Importantly, cheese intake also failed to prevent the reduction in hippocampal 5-HT in PS^#^ rats. We speculate that freezing in PS rats may be a marker of depressive behavior primarily associated with low levels of 5-HT. The positive correlation between freezing and hippocampal 5HIAA concentration, the main 5-HT metabolite, supports our hypothesis.

It is worth noting that the primary effect of resveratrol is its antioxidant action, a characteristic shared with other dietary antioxidants such as tannins and vitamin E [47], which are also present as supplements in cheese and may contribute to its properties [48]. Moreover, apart from antioxidants, cheese also contains vitamin D and calcium due to supplementation.

Cheese supplementation with RES led to a decrease in 5-HT turnover, which is consistent with reports on RES downregulating the serotonin transporter protein (SERT) similar to SSRIs [28]. In half of the RES^+^ rats, freezing behavior disappeared, while the development of anxiety-like behavior was prevented. The other part of RES^+^ rats exhibited freezing behavior with anxiolytic behavior. The reciprocal relationship between anxiety-like behavior and freezing was indicated in Freezing^+^ rats. A recent study by Stairs et al. also reported similar findings, where environmental enrichment increased cue-dependent freezing and behavioral despair but decreased anxiety-like behavior in PTSD rats [49].

Interestingly, Freezing^+^ rats had the lowest BDNF mRNA content among stressed rats. A considerable body of evidence suggests that freezing is associated with fear response, and certain flavonoids with pharmacological activities, such as myricetin, inhibit fear response via activation of the BDNF-ERK signaling pathway, as reported recently [50]. In general, cheese supplementation with RES further promoted the protective effects of cheese intake by inhibiting 5-HT turnover. However, some of the PS^#^ rats happened to be resistant to the protective effects of RES supplementation due to the reduction in BDNF expression.

The main limitations of this study are associated with the high metabolic rate of resveratrol, particularly in the liver. This property of resveratrol has posed significant challenges in both its pharmacodynamics and pharmacokinetics analyses. Initially, we aimed to establish a correlation between behavioral and neurochemical changes and resveratrol blood levels. However, we did not detect the presence of resveratrol and its metabolites such as resveratrol glucuronide, resveratrol sulfate, and dihydro resveratrol in the plasma. This might be linked to the significant interval between the last feeding and euthanasia that was performed the next day after LCI was finalized. The existence of this barrier prompted us to conduct additional tests, where resveratrol was intraperitoneally injected at a dose of 100 mg/kg two hours prior to euthanizing both control and stressed animals. The results of these trials are currently being processed and will be presented in a separate paper.

In this study, we investigated the potential of dietary resveratrol supplementation to address the behavioral and neurochemical abnormalities associated with PTSD. This represents a significant pioneering aspect of the results presented here, as previous studies have primarily regarded resveratrol solely as a treatment for PTSD. Notably, protective effects of resveratrol treatment have been observed in stress-restress models of PTSD. In our study, resveratrol was shown to mitigate the development of time-dependent sensitization (TDS) to events that serve as reminders of past traumatic experiences. We hypothesize that the development of TDS plays a pivotal role in the pathogenesis of PTSD.

This realization has spurred us to explore further avenues of research aimed at gaining a deeper understanding of the protective properties of resveratrol. In our future investigations, we will primarily focus on testing the hypothesis that PTSD arises from disruptions in gene network functionality. A gene network encompasses a collection of genes working in coordination [51]. It is conceivable that PTSD might be attributed to dysregulations in gene network interactions, leading to perturbed interactions among monoamine neurotransmitters across various brain regions. Importantly, the effects of neurotransmitters, such as dopamine (DA), are modulated by trace amines like tyramine, which are abundant in cheese. Thus, PTSD could be conceptualized as a “genetic network disorder”. To explore these notions, we intend to employ bioinformatics, genomics, transcriptomics, and metabolomics approaches. As a result, our forthcoming research on the underlying mechanisms of resveratrol’s protective effects will inherently possess an interdisciplinary character.

## 4. Materials and Methods

In the current study, we utilized a well-validated model of predator stress to experimentally model PTSD. This predator stress (PS) model is relevant to the selective fear response of rodents to a predator and its scent, and it closely mimics acute stress and PTSD in combatants [20,47]. The rats were housed in individually ventilated, standard cages (3–4 rats/cage) and received a high-quality rat chow (Beaphar Care Plus Rat Food, Raalte, The Netherlands,) and tap water ad libitum. The temperature (22–25 °C) and humidity (55%) of the vivarium were controlled, and a 12:12 h light–dark cycle was maintained with lights on between 7:00 and 19:00.

### 4.1. Experimental Animals

Two experimental protocols were performed. In protocol #1, the PS paradigm was utilized to create PS rats. In protocol #2, both the PS and limited cheese intake (LCI) paradigms were combined to create PS^#^ rats.

#### 4.1.1. Predator Stress (PS)

Male Wistar rats (235 ± 12 g, *n* = 30) were housed in groups of 10 rats per cage and provided with standard rat chow and water ad libitum. After an acclimation period, the rats were randomly divided into two groups: a control group (*n* = 10) and an experimental group (*n* = 20) that were exposed to cat urine scent for 15 min daily for 10 days. Subsequently, the rats were allowed to rest for 14 days under stress-free conditions to allow the experimental PTSD to develop [24]. During this time, one rat lost weight and was subsequently excluded from the study. Control rats were housed under similar conditions for the same duration but were not subjected to stress.

#### 4.1.2. Limited Cheese Intake (LCI) Paradigm

The details of cheese manufactory as well as dynamics of changes in the content of resveratrol in the fortified semihard cheese during the ripening period were recorded and can be found in the Appendix A.

Rats were provided with free access to normal chow and water. Twice daily, at 10:00 a.m. and 4:30 p.m., every rat was given an additional piece of cheese 2 g in weight and containing approximately 0.045 kcal per gr of cheese. This amount of cheese was calorically matched to that provided in the twice-daily limited sucrose paradigm and consisted of approximately 33% fat, 24% protein, and 5% carbohydrate by weight, as indicated on the food package labeling. The feeding session began by placing the cheese on the cage floor, and any remaining cheese was removed and weighed after 30 min to determine the amount consumed as a “snack”. Control rats received a small piece of normal chow pre-weighed to provide an equivalent number of calories (0.045 kcal per gr) [35]. In the control group and PS^#^ rats, the LCI paradigm involved the use of control semihard cheese RES^+^ rats received semihard cheese enriched with encapsulated resveratrol) in dose 50 mg/kg. For the enrichment process, 130 mg of resveratrol was added per liter of milk (Appendix A). The encapsulation of resveratrol in liposomes was prepared using lipid components such as phosphatidylcholine, palmitic acid choline ester, and sugar. Liposomes were obtained through a modified method as follows: 650 mg of resveratrol was dissolved in 20 mL of ethanol with continuous stirring on a magnetic stirrer for 10 min. Then, 250 mg of phosphatidylcholine and 40 mg of cholesterol were weighed and added to the astaxanthin solution. The mixture was transferred to a round-bottom flask for evaporation, which was carried out on a rotary evaporator without heating the water bath. The resulting film was dried in a desiccator. Subsequently, hydration was performed with 25 mL of 1% sucrose solution with gentle stirring and heating in a water bath (Appendix A).

The resulting liposomal suspension was a brightly colorless and quite homogeneous mixture. Subsequently, this liposomal suspension was added to the prepared milk (500 mL) and mixed with the rest of the milk in the cheese maker (Appendix A). The subsequent steps followed the method used for obtaining control cheese. As a result, a good clot was obtained during the cheese-making process.

To determine the stability of resveratrol during cheese ripening, HPLC analysis was performed. The dynamics of changes in the content of resveratrol in the fortified semihard cheese during the ripening period were recorded and can be found in the Appendix A.

### 4.2. Behavioral Testing

The anxiety levels of all rats were assessed using the elevated plus maze (EPM) test, conducted with the standard EPM apparatus TS0502-R3 (OpenScience, Moscow, Russia). The total duration of the test was 10 min. Control and experimental rats were tested together in a blind fashion. The behavior of rats in the EPM was recorded and tracked using a SMART video system and analyzed with SMART 3.0 software. Various parameters were measured, including the number of entries into the open and closed arms of the EPM, and the time spent in the open and closed arms. Based on these measurements, the anxiety index (AI) was calculated using the formula [52,53,54]: AI = 1 − {[(time in open arms/Σ time on maze) + (number of entries into open arms/Σ number of all entries)]/2}. The AI discriminant of 0.8 was determined based on the AI distribution of naive rats measured in preliminary experiments for this study and historical AIs of control rats, as previously reported [25]. Additionally, the number of freezing acts in the EPM was also recorded.

### 4.3. Measurement of Hippocampal Concentrations of DA and Its Metabolites

Hippocampal tissue was homogenized in 0.1 M perchloric acid, followed by centrifugation at 7000× *g* for 15 min at 40 °C. The supernatants were then filtered through a syringe with a Whatman filter containing 0.2 µm pores (MilliporeSigma, Burlington, VT, USA) before undergoing high-performance liquid chromatography (HPLC). HPLC was performed under isocratic conditions with electrochemical detection on a Hypersil BDS C18 (250 × 4.6 mm, 5 µm) (Thermo Fisher Scientific, Waltham, MA, USA) reversed-phase column. The mobile phase consisted of a 75 mM phosphate buffer containing 2 mM citric acid, 0.1 mM octanesulphonic acid, and 15% (*v*/*v*) acetonitrile (pH 4.6).

Electrochemical detection was performed with a glassy carbon electrode at +780 mV. The final concentration of monoamines and their metabolites in a tissue sample was expressed as pg/mg tissue using an external calibration curve, as described in our earlier work [20]. The turnover of 5-HT was calculated as 5-HT turnover = [5-HIAA]/[5-HT], and the turnover of dopamine was calculated as dopamine turnover = ([HVA] + [DOPAC])/[dopamine], following the method of Slotkin et al. [55].

### 4.4. Measurement of mRNAs

#### 4.4.1. RNA Isolation

Total RNA was isolated from the hippocampus using TRIzol Reagent (Invitrogen, Oxford, UK). RNA concentrations were measured using a NanoDrop 2000 spectrophotometer (Thermo Fisher Scientific, Waltham, MA, USA) following the standard procedure. The purity of RNA samples was verified by confirming that each had an optical density ratio A260/A280 > 1.8. To ensure the integrity of the samples, the 18S/28S RNA ratio was analyzed after electrophoresis in a 1.4% agarose gel.

#### 4.4.2. cDNA Synthesis and Real-Time RT–PCR

Two μg of total RNA was used for cDNA synthesis using high-capacity cDNA reverse transcription kits (Applied Biosystems/Thermo Fisher Scientific, Waltham, MA, USA). Quantitative real-time RT–PCR was performed using Evrogen 5× qPCR mix–HS SYBR (Evrogen, Moscow, Russia). Primers were designed with the Primer-BLAST software version Primer 3 Plus (National Centre for Biotechnology Information, Boston, MA, USA), and the primer sequences are presented in Table 2.

The PCR parameters were as follows: initial denaturation (one cycle at 95 °C for 15 min); 40 cycles of denaturation, amplification, and quantification (95 °C for 15 s, annealing temperature for 30 s, and 72 °C for 5 s); and the melting curve (starting at 65 °C and gradually increasing to 95 °C). Peptidyl prolyl isomerase A (PPI) mRNA was used as an internal control. The ΔΔCt method was used to determine the fold increase in genes relative to the control group. The results are presented as bar charts. Each value was derived from 2 independent PCR replicates for each cDNA sample, obtained from 5 animals.

### 4.5. Evaluation of MAO-A and MAO-B Activity

Hippocampal MAO-A/MAO-B activity was evaluated in tissue homogenates following the method of Tipton et al. [56]. Brain tissue homogenates were preincubated with 100 µL of 0.5 µM *L*-deprenyl, a selective inhibitor of MAO-B, for 60 min at 37 °C. Afterward, a specific MAO-A substrate, 5-hydroxytryptamine creatinine sulfate (4 mM), was added. For the inhibition of MAO-B activity, 100 µL of 1 µM clorgyline was added to 1 mL of mitochondrial suspension containing MAO in the membrane-bound form and incubated for 60 min at 37 °C. For the evaluation of brain MAO-B activity, benzylamine hydrochloride was used as a substrate. MAO activity was measured spectrophotometrically at 278 nm and expressed as nM serotonin/mg protein/min (MAO-A) or nM benzylamine hydrochloride/mg protein/min (MAO-B).

### 4.6. Data Analyses

Data were analyzed using SPSS 24 (IBM, New York, NY, USA), STATISTICA 10.0 (StatSoft, Tulsa, OK, USA), RStudio 3.3.0+ (RStudio, Boston, MA, USA), and Excel v2308 (Microsoft, Redmond, WA, USA). The normality of data distributions was tested using the Shapiro-Wilk procedure. Data are presented as mean ± SEM or median (25th–75th percentile). Normally distributed data were analyzed with a parametric one-factor ANOVA followed by Tukey’s post hoc tests to compare all outcome measures between respective groups. Non-normally distributed data were analyzed using a nonparametric Mann–Whitney test.

## 5. Conclusions

In conclusion, our study reveals important findings regarding the impact of predator stress (PS) and limited cheese intake (LCI) on anxiety-like behavior and hippocampal monoamines metabolism in rats. PS exposure resulted in the development of anxiety-like behavior, accompanied by alterations in hippocampal dopamine and serotonin concentrations, as well as increased 5-HT turnover rate. However, when rats were subjected to LCI, the anxiety-like response was replaced by freezing behavior, which was associated with a significant increase in hippocampal DA concentration and a decrease in MAO-A/MAO-B activities and gene expression. Additionally, genes related to DA metabolism and brain-derived neurotrophic factor (BDNF) were upregulated in response to LCI.

However, LCI was unable to reverse the increase in 5-HT turnover. In contrast, resveratrol supplementation in cheese led to a decrease in hippocampal 5-HT concentration, similar to the effects of selective serotonin reuptake Inhibitors. The combination of LCI and RES supplementation showed promising effects, optimizing the impact of the LCI paradigm on anxiety-like behavior.

Overall, our findings suggest that limited cheese intake could be considered a potential nutraceutical approach for supporting PTSD treatment. The results highlight the intricate interplay between dietary factors and stress-related behavior, offering valuable insights for future studies in this area.

## Figures and Tables

**Figure 1 ijms-24-14343-f001:**
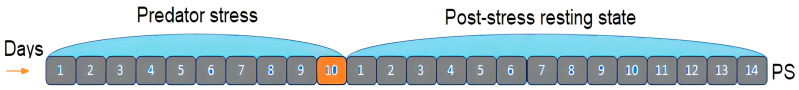
The timeline of protocol #1. Grey color represents stressed rats on the usual food ration. Orange color represents the last day of predator stress exposure.

**Figure 2 ijms-24-14343-f002:**
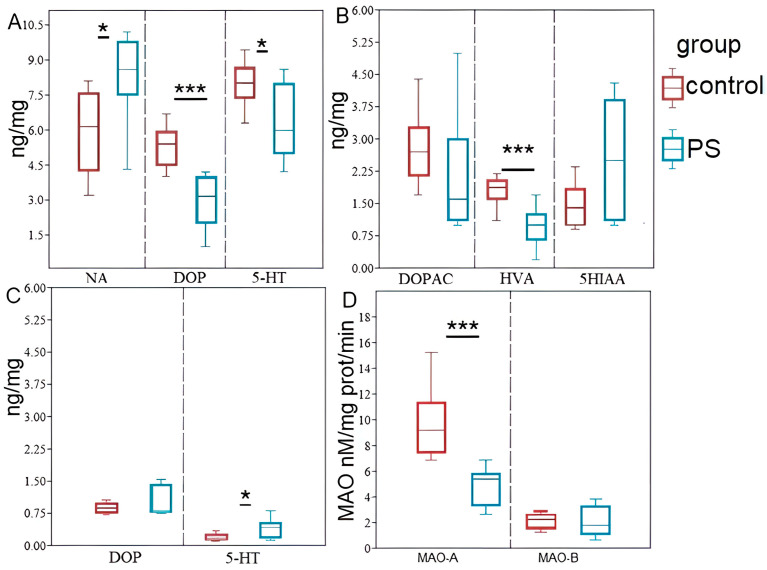
The hippocampal levels of monoamines their metabolites (DOPAC, HVA, 5-HIAA) (**B**), turnover rates (**C**) and MAO activities (**D**) in PS rats; Panel (**A**) NA, DA and 5-HT concentrations, Panel (**B**) DOPAC, HVA and 5-HIAA concentrations, Panel (**C**) DOP and 5-HT turnover rates, Panel (**D**) MAO-A/MAO-B activities. Values are presented as means ± SD. Statistical significance levels: * *p* < 0.05; *** *p* < 0.001 indicate differences from the control group.

**Figure 3 ijms-24-14343-f003:**
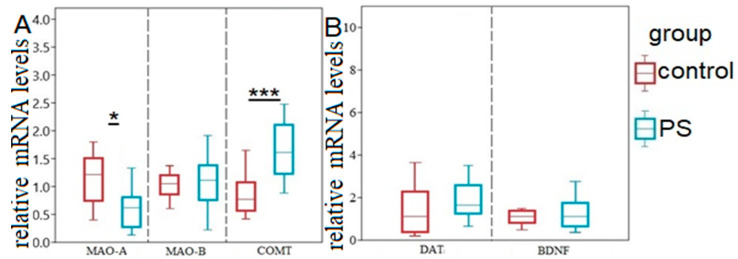
Gene expression of hippocampal metabolic enzymes, DAT, and BDNF in PS panel rats. Panel (**A**) relative mRNA levels of MAO-A, MAO-B, COMT. Panel (**B**) relative mRNA levels of DAT and BDNF. Values are presented as means ± SD. Statistical significance levels: * *p* < 0.05; *** *p* < 0.001 indicate differences from the control group.

**Figure 4 ijms-24-14343-f004:**
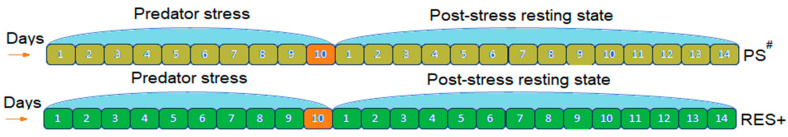
The timeline of Protocol#2 LCI paradigm in PS rats (PS^#^ group upper panel). Khaki color represents rats fed with cheese without addition of resveratrol; intake cheese with resveratrol supplementation in PS rats (RES^+^ group; lower panel). Green color represents rats fed with resveratrol-enriched cheese; control group had LCI without PS exposures. Orange color represents the last day of predator stress exposure.

**Figure 5 ijms-24-14343-f005:**
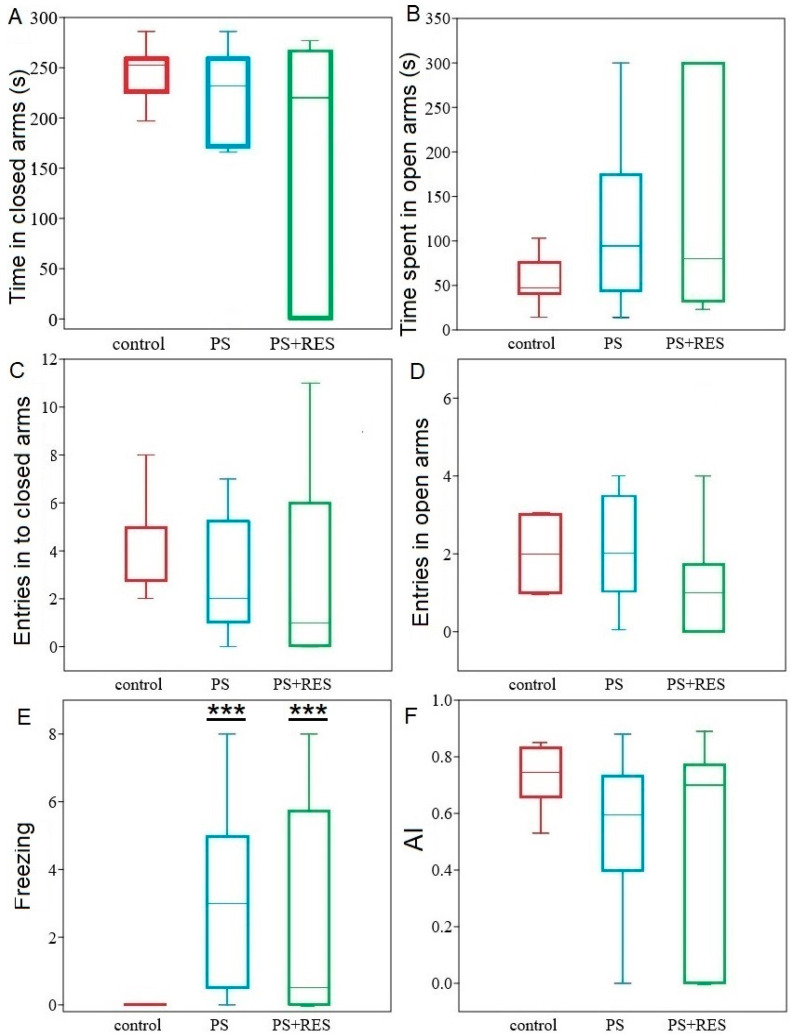
Behavioral activity in the EPM was evaluated in control, PS^#^ rats and PS + RES rats: Panel (**A**) time in closed arms, Panel (**B**), time spent in open arms, Panel (**C**) entries in to closed arms, Panel (**D**) entries in open arms, Panel (**E**) freezing, and Panel (**F**) AI. The values are presented as means ± SD. Significant differences from the control group are indicated by *** for *p* < 0.0001.

**Figure 6 ijms-24-14343-f006:**
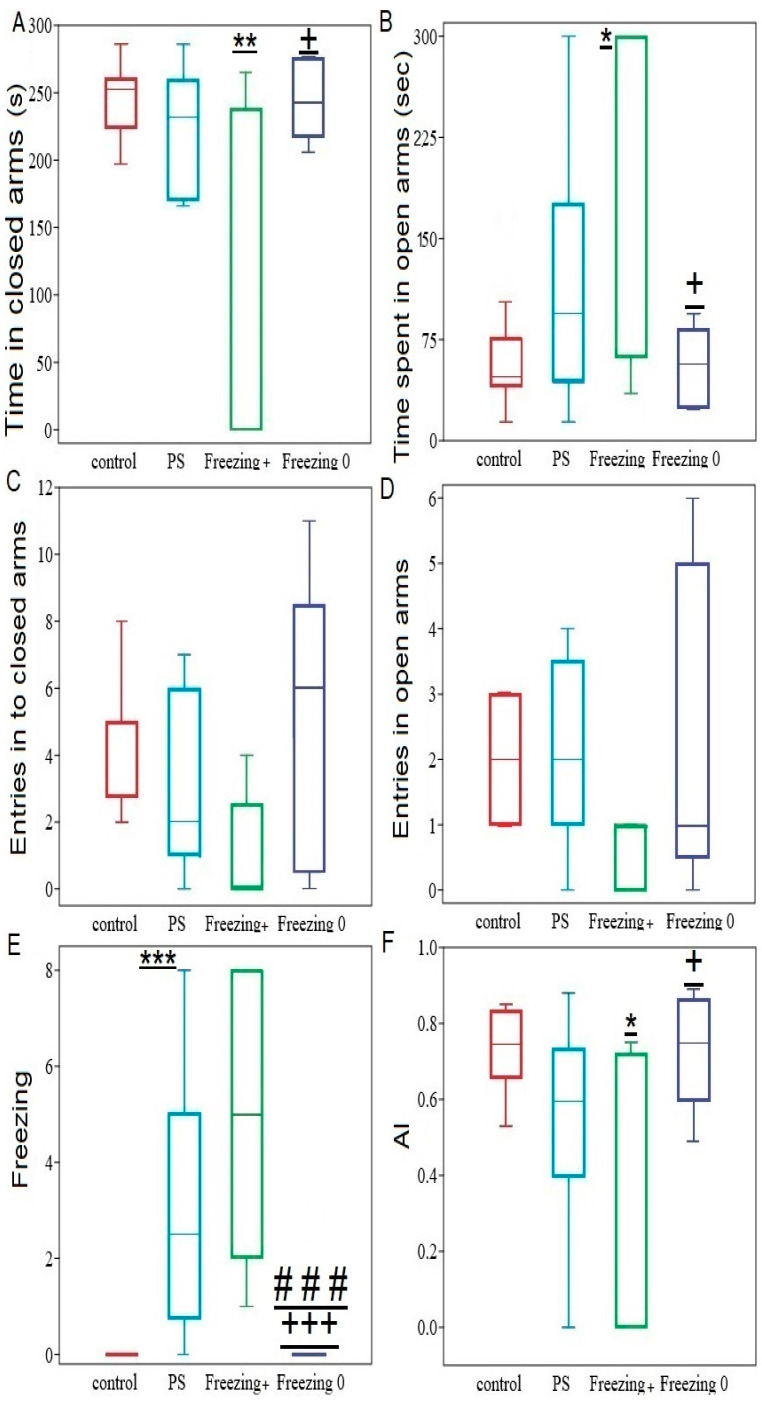
Behavioral activity in the EPM test among PS^#^, Freezing^+^, and Freezing^0^: Panel (**A**) time in closed arms, Panel (**B**) time spent in open arms, Panel (**C**) entries in to closed arms, Panel (**D**) entries in open arms, Panel (**E**) freezing, and Panel (**F**) AI. The values are represented as means ± SD. Significance levels for differences with the control group are indicated as follows: * *p* < 0.05, ** *p* < 0.01, *** *p* < 0.0001. Additionally, differences with the PS^#^ rats are marked with ### (*p* < 0.01); differences between freezing+ and freezing^0^ phenotypes + *p* < 0.05; +++ *p* < 0.0001.

**Figure 7 ijms-24-14343-f007:**
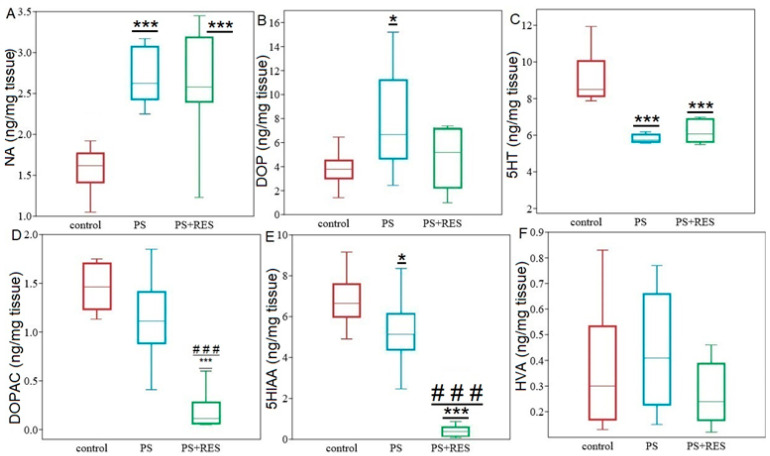
Hippocampal concentrations of monoamines (and their metabolites (DOPAC (D), in PS^#^ and PS + RES rats. Panel (**A**) NA concentration, Panel (**B**) DA concentration, Panel (**C**) 5-HTconcentration, Panel (**D**) DOPAC concentration, Panel (**E**) 5 HIAA concentration, Panel (**F**) HVA concentration Values are presented as means ± SD. *; *** indicate differences with the control group; * *p* < 0.05; *** *p* < 0.0001. ### indicate differences with PS^#^ rats; ### *p* < 0.01.

**Figure 8 ijms-24-14343-f008:**
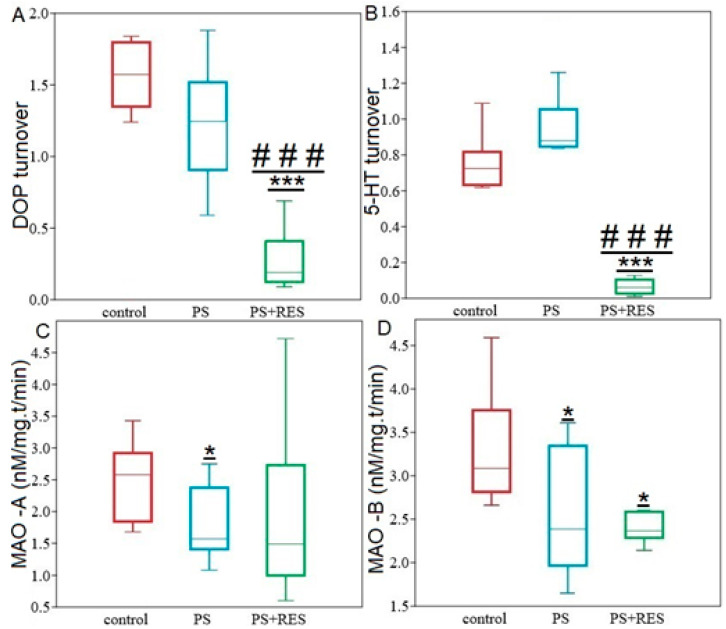
Hippocampal monoamines turnover and MAO activities in PS^#^ and PS+RES rats. Panel (**A**) DA turnover rates, Panel (**B**) 5–HT turnover rates, Panel (**C**) MAO-B activity, Panel (**D**) MAO-A activities. Values are presented as means ± SD. *; *** indicate significant differences compared to the control group; * *p* < 0.05; *** *p* < 0.0001. ### indicate significant differences compared to PS^#^ rats; ### *p* < 0.01.

**Figure 9 ijms-24-14343-f009:**
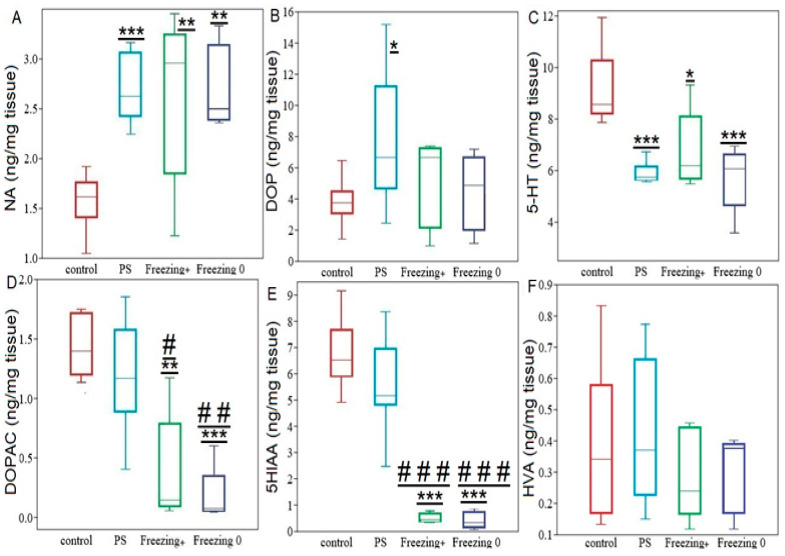
Hippocampal monoamines, and their metabolites levels in PS^#^, Freezing^+^, and Freezing^0^ rats. Panel (**A**) NA concentration, Panel (**B**) DA concentration, Panel (**C**) 5-HTconcentration, Panel (**D**) DOPAC concentration, Panel (**E**) 5 HIAA concentration, Panel (**F**) HVA concentration. Values are expressed as means ± SD. *; **; *** indicate differences compared to the control group; * *p* < 0.05; ** *p* < 0.01; *** *p* < 0.0001. #; ##, ### indicate differences compared to PS^#^ rats; # *p* < 0.05; ## *p* < 0.01, ### *p* < 0.001.

**Figure 10 ijms-24-14343-f010:**
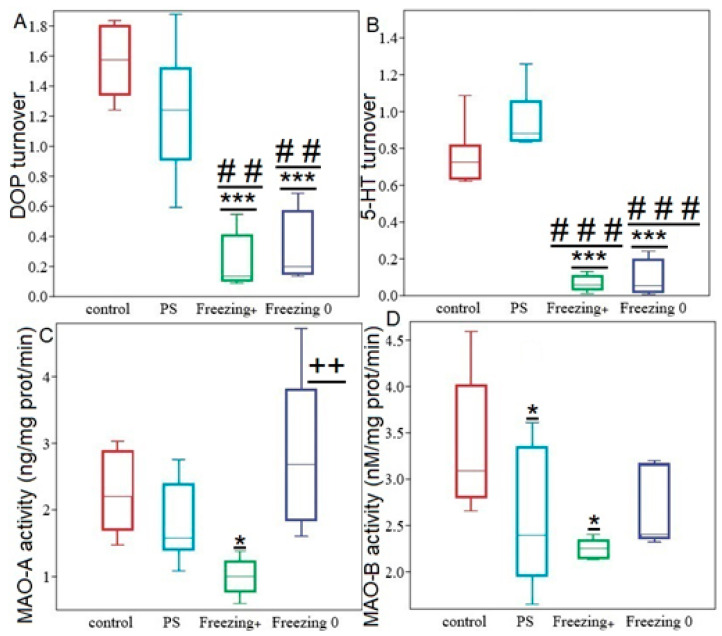
Hippocampal monoamins turnover r, MAO- activities in PS^#^, Freezing+, and Freezing0 rats. Panel (**A**) DOP turnover rates. Panel (**B**) 5-HT turnover rates. Panel (**C**) MAO-A activity. Panel (**D**) MAO-B activity Values are expressed as means ± SD. *; *** indicate significant differences compared to the control group; * *p* < 0.05; *** *p* < 0.0001. ##, ### indicate significant differences compared to PS^#^ rats; ## *p* < 0.01, ### *p* < 0.0001. ++ indicates significant differences between Freezing^+^ and Freezing^0^ rats; ++ *p* < 0.01.

**Figure 11 ijms-24-14343-f011:**
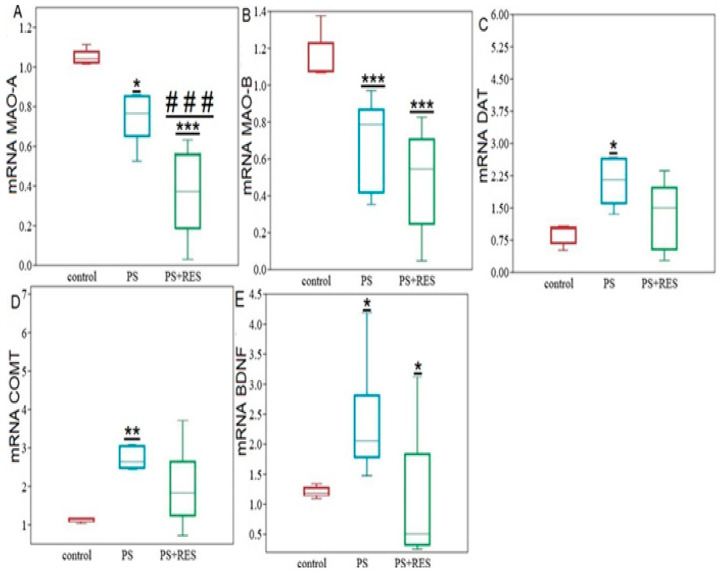
Gene expression of hippocampal metabolic enzymes, DAT, and BDNF in PS^#^ and PS + RES rats: Panel (**A**) mRNA MAO-A relative levels, Panel (**B**) mRNA MAO-B relative levels, Panel (**C**) mRNA DAT relative levels. Panel (**D**) mRNA COMT relative levels, Panel (**E**) mRNA BDNF relative levels. Values are means ± SD. *; **; *** indicate significant differences with the control group; * *p* < 0.05; ** *p* < 0.01; *** *p* < 0.001. ### indicate significant differences with PS^#^ rats; ### *p* < 0.0001.

**Figure 12 ijms-24-14343-f012:**
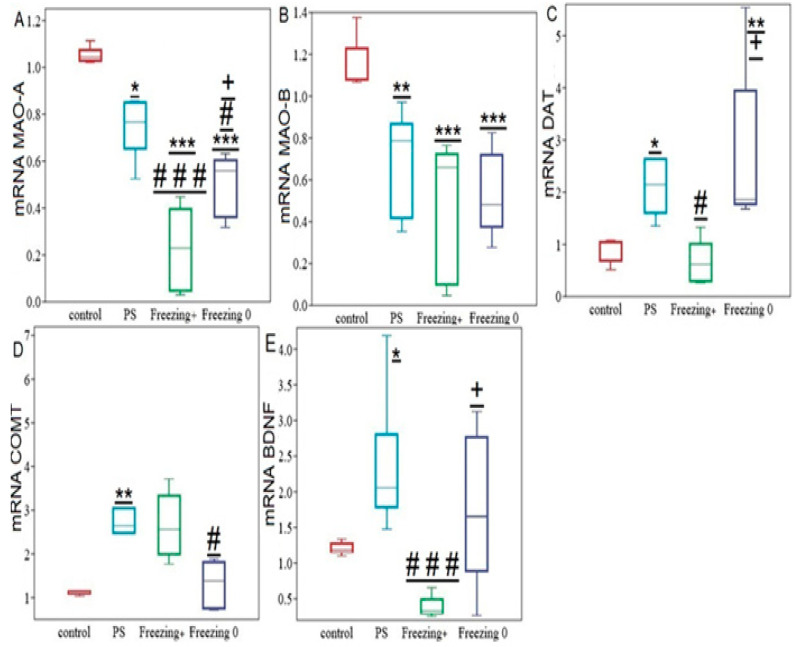
Gene expression of hippocampal metabolic enzymes, DAT, and BDNF in PS^#^, Freezing^+^, and Freezing^0^ rats: Panel (**A**) mRNA MAO-A relative levels, Panel (**B**) mRNA MAO-B relative levels, Panel (**C**) mRNA DAT relative levels. Panel (**D**), mRNA COMT relative levels, Panel (**E**) mRNA BDNF relative levels. Values are expressed as means ± SD. Significant differences with the control group are indicated by * for * *p* < 0.05, ** for ** *p* < 0.01, and *** for *** *p* < 0.0001. Differences with the PS^#^ rats are denoted by # for # *p* < 0.05 and ### for ### *p* < 0.01, while differences between Freezing^+^ and Freezing^0^ rats are represented by + for + *p* < 0.05.

**Figure 13 ijms-24-14343-f013:**
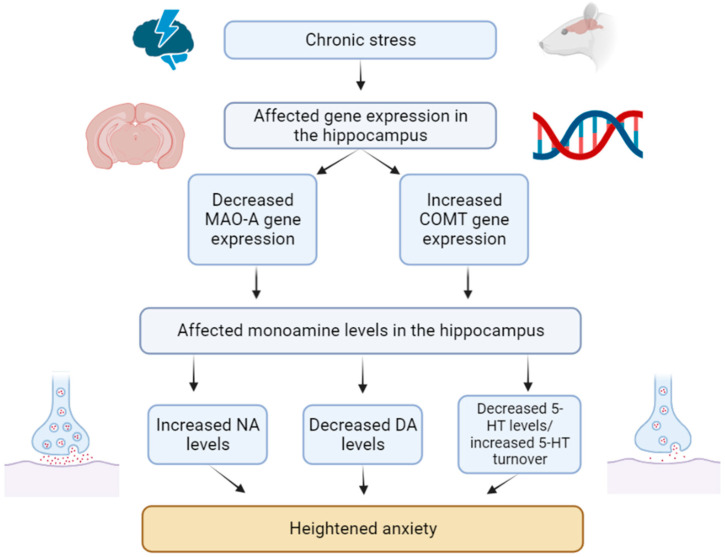
General scheme of chronic stress impact on monoamine neurotransmission in the hippocampus and its relation to anxiety phenotype in PS-exposed rats.

**Figure 14 ijms-24-14343-f014:**
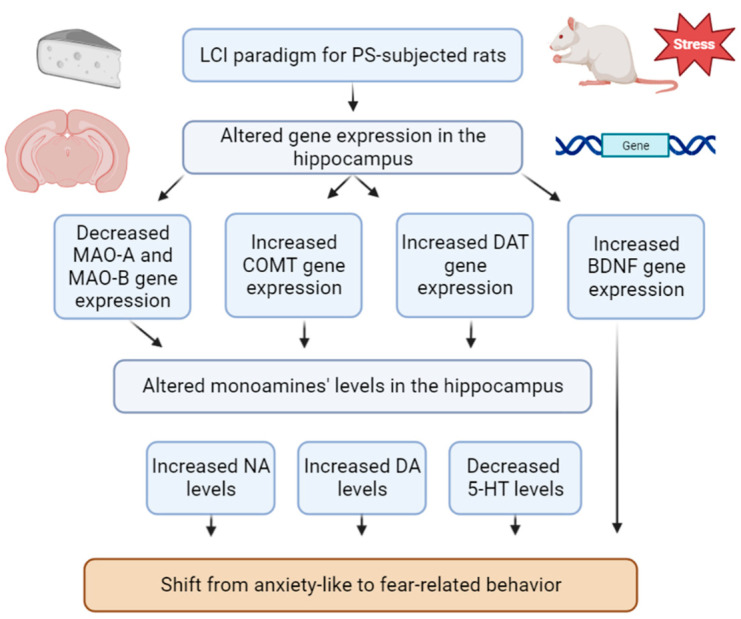
Conceptual scheme of the limited cheese intake paradigm in PS-subjected rats.

**Figure 15 ijms-24-14343-f015:**
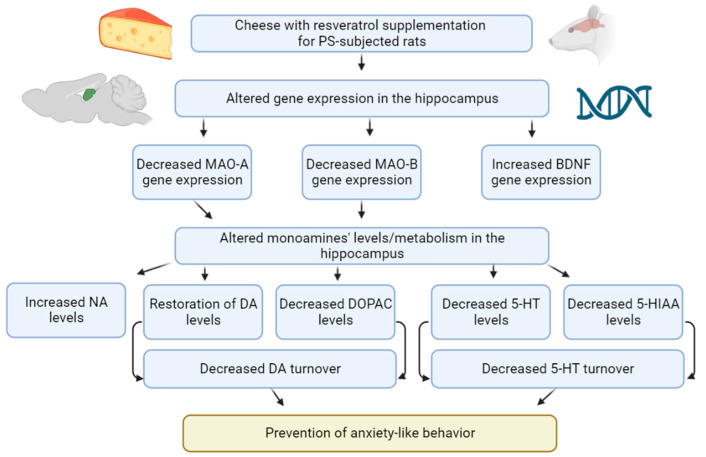
Cheese diet with resveratrol supplementation in PS-exposed rats.

**Figure 16 ijms-24-14343-f016:**
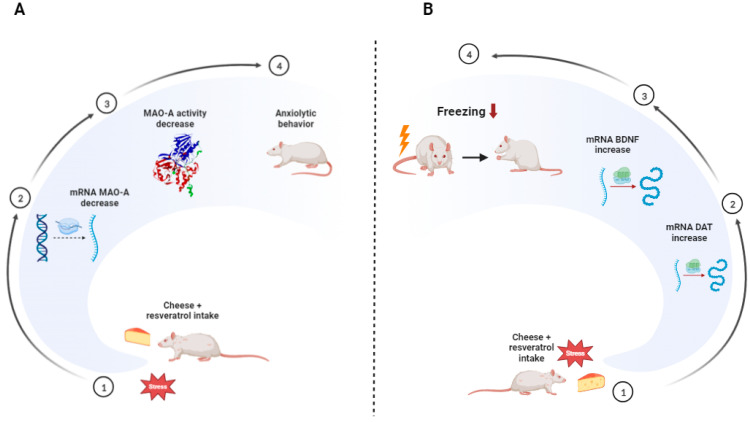
Anxiolytic behavior and reduction in freezing observed in PS-subjected rats fed a diet of cheese supplemented with resveratrol, attributed to altered expression of MAO-A, DAT, and BDNF. Panel (**A**) Freezing+ rats. 1—Predator Stress, 2—mRNA MAO levels, 3—MAO-A activity, 4—Anxiolytic behavior. Panel (**B**) Freezing^0^ rats. 1—Predator Stress, 2—mRNA DAT levels. 3—mRNA BDNF levels, 4—Freezing behavior.

**Table 1 ijms-24-14343-t001:** Behavioral activity of control and PS-exposed rats in the elevated plus maze (EPM) test.

Variables	Control (*n* = 10)	PS (*n* = 15)
time spent in open arms (s)	51.85 (46.1; 58.3)	34.25 * (22.5; 41.4)
time spent in closed arms (s)	248.14 (253.9; 269.7)	265.75 * (258.6;277.5)
entries into open arms (s)	5.0 (2; 7)	2.0 * (1;4)
entries into closed arms (s)	7.0 (5; 11)	10.0 * (5; 12)
anxiety index (AI)	0.706 (0.64; 0.73)	0.86 (0.8; 0.91)
freezing (number acts in EPM)	1.0 (0; 1)	2.1 (0; 4)

* Data are median (first quartile; third quartile) Statistical significance levels: * *p* < 0.05; indicate differences from the control group.

**Table 2 ijms-24-14343-t002:** Primer Sequences for Real-Time RT-PCR.

Name of the Gene	Primer Sequence 5′→3′	Annealing Temperature, °C
PPI	F GATTTGGCTATAAGGGTTC R GTTGTCCACAGTCGGAGA	60
MAO-A	F GCCAGGAACGGAAATTTGTA R TCTCAGGTGGAAGCTCTGGT	64
MAO-B	F TGGGCCAAGAGATTCCCAGTGATG R AGAGTGTGGCAATCTGCTTTGTAG	60
Comt	F CTGGAGGCCATCGACACCTA R AGTAAGCTCCCAGCTCCAGCA	60
BDNF	F GAAAGTCCCGGTATCAAAAG R CGCCAGCCAATTCTCTTTTTG	60
DAT	F TTGGGTTTGGAGTGCTGATTGC R AGAAGACGACGAAGCCAGAGG	55

## Data Availability

The data presented in this study are available in article.

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
