# Peer review of "Limited Cheese Intake Paradigm Replaces Patterns of Behavioral Disorders in Experimental PTSD: Focus on Resveratrol Supplementation"

_ijms, 2023, doi:10.3390/ijms241814343_

Round 1

Reviewer 1 Report

I have reviewed the manuscript titled "Limited Cheese Intake Paradigm Replaces Patterns of Behavioral Disorders in Experimental PTSD: Focus on Resveratrol Supplementation." I must commend the authors on their innovative approach and insightful findings in addressing the behavioral disorders associated with PTSD.

Your use of a limited cheese intake paradigm to modulate behavioral patterns is a unique and intriguing concept. The idea of dietary interventions, such as resveratrol supplementation, as potential adjunct therapies for PTSD is particularly noteworthy. By exploring alternative treatments beyond traditional pharmaceutical approaches, your study opens up new avenues for research and therapeutic development.

The inclusion of resveratrol, a compound known for its potential health benefits, adds an interesting dimension to your work. Your study's focus on its effects on behavioral disorders in the context of PTSD could have significant implications for both clinical practice and further research in the field of mental health.

However, I would like to suggest a few points for consideration:

Discussion of Limitations: Every study has limitations, and discussing the limitations of your research is crucial for contextualizing the significance of your findings. Addressing potential confounding variables, limitations in the animal model used, and the generalizability of your results to human populations would contribute to a more well-rounded discussion.

Comparison with Existing Literature: It would be helpful to compare and contrast your findings with existing literature on both PTSD models and the effects of resveratrol supplementation on mental health-related outcomes. This would provide a broader context for your study and highlight the novel contributions you've made.

Overall, the manuscript presents a captivating exploration of an unconventional approach to managing PTSD-related behavioral disorders.

Author Response

We extend our gratitude to the reviewer for their comprehensive analysis of our article and for providing a positive evaluation. We have carefully considered the reviewer's suggestions and incorporated the suggested additions into the text.

Discussion of Limitations: Every study has limitations, and discussing the limitations of your research is crucial for contextualizing the significance of your findings. Addressing potential confounding variables, limitations in the animal model used, and the generalizability of your results to human populations would contribute to a more well-rounded discussion.

Answer: A new paragraph discussing the limitations of our research has been incorporated into the Discussion section.

Action: The main limitations of this study are associated with the high metabolic rate of resveratrol, particularly in the liver. This property of resveratrol has posed significant challenges in both its pharmacodynamics and pharmacokinetics analyses. Initially, we aimed to establish a correlation between behavioral and neurochemical changes and resveratrol blood levels. However, we did not detect the presence of resveratrol in the plasma. The existence of this barrier prompted us to conduct additional tests, where resveratrol was intraperitoneally injected at a dose of 100 mg/kg two hours prior to euthanizing both control and stressed animals. The results of these trials are currently being processed and will be presented in a separate paper.

Comparison with Existing Literature: It would be helpful to compare and contrast your findings with existing literature on both PTSD models and the effects of resveratrol supplementation on mental health-related outcomes. This would provide a broader context for your study and highlight the novel contributions you've made.

Answer: A new paragraph has been incorporated into the Discussion section.

Action: In this study, we investigated the potential of dietary resveratrol supplementation to address the behavioral and neurochemical abnormalities associated with PTSD. This represents a significant pioneering aspect of the results presented here, as previous studies have primarily regarded resveratrol solely as a treatment for PTSD. Notably, protective effects of resveratrol treatment have been observed in stress-restress models of PTSD. In our study, resveratrol was shown to mitigate the development of time-dependent sensitization (TDS) to events that serve as reminders of past traumatic experiences. We hypothesize that the development of TDS plays a pivotal role in the pathogenesis of PTSD.

This realization has spurred us to explore further avenues of research aimed at gaining a deeper understanding of the protective properties of resveratrol. In our future investigations, we will primarily focus on testing the hypothesis that PTSD arises from disruptions in gene network functionality. A gene network encompasses a collection of genes working in coordination [Chen S, Chen X, Su H, Guo M, Liu H. Advances in Synthetic-Biology-Based Whole-Cell Biosensors: Principles, Genetic Modules, and Applications in Food Safety. Int J Mol Sci. 2023 Apr 28;24(9):7989. doi: 10.3390/ijms24097989. PMID: 37175695; PMCID: PMC10178329.]. It is conceivable that PTSD might be attributed to dysregulations in gene network interactions, leading to perturbed interactions among monoamine neurotransmitters across various brain regions. Importantly, the effects of neurotransmitters, such as dopamine (DA), are modulated by trace amines like tyramine, which are abundant in cheese. Thus, PTSD could be conceptualized as a "genetic network disorder." To explore these notions, we intend to employ bioinformatics, genomics, transcriptomics, and metabolomics approaches. As a result, our forthcoming research on the underlying mechanisms of resveratrol's protective effects will inherently possess an interdisciplinary character.

Reviewer 2 Report

The article is very well-written and provides the scientific relevance of the findings based on proper research methodology. I highly recommend this article for publication. The quality of data and the depiction meets the standards.

I only have two questions which may be interesting for other readers as well:

1. Can you mention other proven or theoretical clinical use of trans-resveratrol in the introduction? (Briefly)

2. What other active substances may have interaction with the limitation of cheese intake?

3. Would we suspect any signifant supplementary resveratrol dosage dependency?

Author Response

We extend our gratitude to the reviewer for their comprehensive analysis of our article and for providing a positive evaluation. We have carefully considered the reviewer's suggestions and incorporated the suggested additions into the text.

Comment 1: Can you mention other proven or theoretical clinical use of trans-resveratrol in the introduction? (Briefly)

Answer: A new paragraph highlighting the established benefits of trans-resveratrol has been added to the Introduction section.

Action: Nowadays, the multifaceted physiological benefits of resveratrol have garnered considerable attention, showcasing its potential to enhance healthspan across diverse experimental models of various behavioral disorders. Clinical trials have also lent support to the ongoing exploration of resveratrol treatment in human contexts [Blanchard OL, Friesenhahn G, Javors MA, Smoliga JM. Development of a lozenge for oral transmucosal delivery of trans-resveratrol in humans: proof of concept. PLoS One. 2014 Feb 26;9(2):e90131. doi: 10.1371/journal.pone.0090131. PMID: 24587240; PMCID: PMC3935991]. Notably, trans-resveratrol supplementation has demonstrated its ability to enhance cognitive function in the elderly [Anton SD, Ebner N, Dzierzewski JM, Zlatar ZZ, Gurka MJ, Dotson VM, Kirton J, Mankowski RT, Marsiske M, Manini TM. Effects of 90 Days of Resveratrol Supplementation on Cognitive Function in Elders: A Pilot Study. J Altern Complement Med. 2018 Jul;24(7):725-732. doi: 10.1089/acm.2017.0398. Epub 2018 Mar 27. PMID: 29583015; PMCID: PMC6065512]. 

Comment 2: What other active substances may have interaction with the limitation of cheese intake?

Answer: A new paragraph highlighting the established benefits of trans-resveratrol has been added to the Introduction section.

Action: It's worth noting that the primary effect of resveratrol is its antioxidant action, a characteristic shared with other dietary antioxidants such as tannins and vitamin E [Politis I, Bizelis I, Tsiaras A, Baldi A. Effect of vitamin E supplementation on neutrophil function, milk composition and plasmin activity in dairy cows in a commercial herd. J Dairy Res. 2004 Aug;71(3):273-8. doi: 10.1017/s002202990400010x. PMID: 15354572], which are also present as supplements in cheese and may contribute to its properties [Santillo A, Ciliberti MG, Ciampi F, Luciano G, Natalello A, Menci R, Caccamo M, Sevi A, Albenzio M. Feeding tannins to dairy cows in different seasons improves the oxidative status of blood plasma and the antioxidant capacity of cheese. J Dairy Sci. 2022 Nov;105(11):8609-8620. doi: 10.3168/jds.2022-22256. Epub 2022 Sep 27. PMID: 36175229]. Moreover, apart from antioxidants, cheese also contains vitamin D and calcium due to supplementation.

Comment 3. Would we suspect any signifant supplementary resveratrol dosage dependency?

Answer: In this study, the concentration of RES supplementation was 0.87±0.11 mg/g, which is relatively low. Thus, it is unlikely that dependencies on such a low concentration would be present.
